# Selective Anthelmintic Treatment in Horses in Sweden Based on Coprological Analyses: Ten-Year Results

**DOI:** 10.3390/ani13172741

**Published:** 2023-08-28

**Authors:** Eva Osterman-Lind, Mia Holmberg, Giulio Grandi

**Affiliations:** 1Department of Microbiology, National Veterinary Institute (SVA), 751 89 Uppsala, Sweden; eva.osterman-lind@sva.se; 2Department of Disease Control and Epidemiology, National Veterinary Institute (SVA), 751 89 Uppsala, Sweden; mia.holmberg@sva.se; 3Department of Biomedical Sciences and Veterinary Public Health, Swedish University of Agricultural Sciences (SLU), 751 89 Uppsala, Sweden

**Keywords:** parasite control, deworming, faecal sample, *Strongylus vulgaris*

## Abstract

**Simple Summary:**

In the past, horses were protected from parasitic infections by routine deworming aimed at keeping harmful worms such as *Strongylus vulgaris* under control. The emergence of drug-resistant worm populations has stimulated the introduction of alternative control strategies to reduce the frequency of antiparasitic treatments. In Sweden, selective treatment is recommended, and deworming should be based on the results of faecal analyses, where individual horses that contribute substantially to pasture contamination are treated. In our study, horses from equestrian premises across Sweden participated in a parasite monitoring programme during the years 2008–2017. Faecal samples were collected in the spring and in the autumn in order to identify which individuals should be dewormed. According to the results obtained from the spring analyses (March to June), the use of antiparasitic drugs could be greatly reduced when selective treatment is chosen instead of routine treatment.

**Abstract:**

In Sweden, routine deworming has been used for several decades; however, to slow down the development of anthelmintic resistance, selective treatment is currently recommended. As part of a monitoring programme, equestrian premises submitted faecal samples to the National Veterinary Institute (SVA) twice per year between 2008 and 2017. Analyses for strongyles (small and large), tapeworms and ascarids, followed by premise-specific advice regarding deworming and parasite control strategies, were provided. In total, 43,330 faecal samples, collected from 26,625 horses on 935 premises in springtime (March to June), were analysed by quantitative or semi-quantitative flotation. Moreover, *Strongylus vulgaris* was detected by larval culture or PCR. Between 4 and 11% of individual horses tested positive for *S. vulgaris* and 3–10% were shedding tapeworm eggs. There were recurrent high and low egg shedders; 75% of horses with *S. vulgaris* appeared to have been recently introduced into the herd; the proportion of *S. vulgaris*-positive premises increased when individual samples rather than pooled samples were used. Based on the results of *S. vulgaris* diagnostics and strongyle egg-shedding level, 59% of the horses did not need to be dewormed.

## 1. Introduction

The vast majority of horses in Sweden (*n* = 355,500 according to Statistics Sweden [1]) are for leisure or sport and the horse density on the grazing areas is usually high, allowing for the accumulation of infectious stages of parasites on the pastures. Larval stages of small strongyles (cyathostomins) and *Strongylus vulgaris*, as well as tapeworms and ascarids, can cause severe, and sometimes life-threatening disease [2]. Strategies to control parasitic infections are thus fundamental if horses are to be protected from disease, especially in premises where young horses are present.

Routine deworming primarily aimed at *S. vulgaris* control/eradication has resulted in the emergence of anthelmintic resistance in both small strongyles and ascarids worldwide [2,3]. To slow down the progression of anthelmintic resistance, selective treatment has been recommended during the past two decades, along with methods such as dung removal and alternate grazing. Dung removal at a rate of twice per week has been shown to be very effective at reducing the number of infective strongyle larvae of horses and donkeys on pastures [4,5]. Although deworming without prior faecal analysis is still permitted in Sweden, the treatment decision is usually based on the results of individual faecal analyses, with 200 strongyle eggs per gram (EPG) often used as the threshold for deworming [6].

In 2006, a parasite-monitoring programme for equestrian premises (with at least eight horses) was initiated and marketed by the National Veterinary Institute (SVA) in Uppsala, Sweden, to enable parasite control based on targeted selective treatment. The aim of this programme is to achieve an effective control of parasites in horse premises, and simultaneously slow-down in the progression of anthelmintic resistance by avoiding unnecessary treatments and encouraging the adoption of various pasture hygiene methods. In line with the SVA’s practices, a veterinarian in the parasitology laboratory communicates the results of the faecal analyses and provides the equestrian premise with specific advice about anthelmintic treatment and pasture management. A practising veterinarian connected to the equestrian premise then evaluates the advice and prescribes the anthelmintics. The laboratory methods used in the programme have been chosen and developed on the basis of diagnosing the most important intestinal parasites (*S. vulgaris*, cyathostomins, *Anoplocephala perfoliata* and *Parascaris* spp.) at a price that horse owners find reasonable.

The aim of this article is to demonstrate how selective treatment can be used in practice to control parasites on Swedish equestrian premises. We also show how diagnostic methods were adapted for the detection of helminths in large-scale diagnostics. Additionally, this paper presents figures representing the occurrence of the most common equine parasites during the years of implementation.

## 2. Materials and Methods

### 2.1. The Monitoring Programme

Faecal samples from individual horses were collected and submitted to the SVA by owners of horses or equestrian premises in the spring and autumn. In connection with the first sampling, information on herd size, type of equestrian premise, horse ages, deworming routines and pasture management practices was obtained. In the spring, analyses were performed according to Section 2.3 to assess the intensity of parasite egg shedding and detect *S. vulgaris* and/or *A. perfoliata*. In the autumn, faecal egg counts (FECs) were carried out. Based on the laboratory results, information from the premise and the previous two years’ faecal analysis results, specific advice was formulated and tailor-made for each equestrian premise. Moreover, a drug efficacy test was recommended whenever at least eight horses were shedding more than 200 EPG/moderate numbers of strongyle eggs in the spring. The minimum number of horses was chosen based on the fact that it is recommended to perform efficacy tests on groups of at least 6 horses [7]. The costs of analyses and the veterinary service were paid for by the equestrian premises. To minimise the risk of introducing *S. vulgaris* and tapeworms, it was advised that newly introduced horses should be dewormed with macrocyclic lactones in combination with praziquantel upon arrival and prior to inclusion in the monitoring programme. A drug efficacy test was also recommended for these individuals.

### 2.2. Data Included

Only the results from analyses of faecal samples submitted in the spring (1 March–30 June) in the years 2008–2017 were included in this study. Information on horse age was recorded from 2009 onwards whenever provided by the owner. Information such as aging of individual horses over time, pasture management on the farms and previous anthelmintic treatments could not be used in the data analyses since they had not been registered in a standardised way in the SVA’s data management system.

### 2.3. Analyses for Helminth Eggs

Faeces were collected by the horse owners and packed according to recommendations from SVA. Samples were then sent to the laboratory by regular mail or delivered by the owners themselves. Upon arrival, the samples were stored at +4 °C until analysis within 1–72 h. An overview of the different detection techniques employed over the years is presented in Table 1.

In 2008–2016, individual samples were analysed using a modified McMaster technique based on 3 g of faeces [8], with a minimum detection level of 50 eggs per gram (EPG). The flotation medium used was a saturated sodium chloride solution with 50% glucose and a specific gravity of 1.280. A centrifugation-based flotation of the faecal suspension left from the counting technique was performed to recover tapeworm eggs, since the McMaster technique alone is not suitable for their diagnosis. Even by adding this step, the detection of tapeworm eggs was based on the examination of about 1 g of faeces only.

Because of tapeworm biology (eggs being excreted clustered in proglottids), the method described above was not considered adequate for a sensitive detection of these parasites. In 2017, the McMaster technique (a dilution counting technique) was replaced with a modified flotation method (a concentration technique) to improve the detection of parasite eggs, especially tapeworm eggs [9]. Briefly, 30 g of faeces was mixed thoroughly with 60 mL tap water and passed through a 150 μm sieve. The sieved content was centrifuged for 10 min at 1000× *g*. Then, the supernatant was discarded, and the pellet was resuspended in a sugar–salt solution and centrifuged for 5 min at 214× *g* with a coverslip on the top. The cover glass was then microscopically examined at 40–100× magnification. The number of nematode eggs was assessed according to a semi-quantitative scale (Table 2). According to an internal comparison between the McMaster egg counting technique and the presently described flotation technique, a moderate number of eggs corresponded to 200–650 EPG (SVA internal data). The presence/absence of *A. perfoliata* eggs was registered. Material recovered from the same microscope slides was used for PCR detection of *S. vulgaris* (see below).

### 2.4. Analyses for Strongylus vulgaris

In 2008–2012, pooled larval cultures were performed to identify equestrian premises infected with *S. vulgaris*. Approximately 40 g of faeces from four horses (10 g from each individual horse) was incubated at 25 °C for 10 days. Strongyle third-stage larvae were collected and microscopically examined to identify larvae of *S. vulgaris* [10]. In 2013–2016, equestrian premises had to indicate on the request form whether they wanted to have samples pooled or analysed individually. In the latter case, approximately a four-times-greater volume of faeces was used from each individual. For all years, larval cultures were performed regardless of EPGs.

In 2017, samples from equestrian premises that requested individual analyses for *S. vulgaris* were analysed by PCR to shorten the response time. The material on the microscope slides from the modified flotation technique (see Section 2.3) was transferred to a test tube. The samples were denaturated at >97 °C for 10 min, and nucleic acid extraction was performed by means of a magnetic bead-based extraction using a Bullet Stool kit (Diasorin, Stillwater, MN, USA) and a Magnatrix 8000+ extraction robot (Magnetic Biosolutions, Stockholm, Sweden). For the extraction, 90 μL of the denaturated sample was added to separate wells on a 1.2 mL square-well storage plate (Thermo-Fisher Scientific, Gothenburg, Sweden, Cat. no. AB-1127) together with 10 μL ≥ 800 U/mL proteinase K (Sigma-Aldrich, Saint Louis, MO, USA), and run in the extraction robot on a protocol modified for using the kit on the Magnatrix 8000+. The modification involved an extra post-lysis wash using a 20% dilution of the lysis buffer provided in the kit, and two washes with wash buffers 1 and 2, respectively. Positive and negative controls, which were also used as controls in the real-time PCR, were included in each extraction. After extraction, the samples were analysed by real-time PCR.

For the real-time PCR, a SsoFast Probes Supermix PCR kit (Bio-Rad, Life Technologies) was used together with the forward primer GTATACATTAAATAGTGTCCCCCATTCTAG, reverse primer GCAAATATCATTAGATTTGATTCTTCCG and probe FAM-TGGATTTATTCTCACTACTTAATTGTTTCGCGAC-BHQ1 [11] (Eurofins MWG Operon, Ebersberg, Germany). A total of 2 μL of extracted nucleic acid and 13 μL mastermix were used per sample. The mastermix comprised 7.5 μL SsoFast Probes Supermix, 0.6 μL forward primer (10 μM), 0.6 μL reverse primer (10 μM), 0.2 μL probe (10 μM) and 4.1 μL nuclease-free H_2_O. PCR was then performed on a 7500 Fast Real-Time PCR System (ABI) as follows: 95 °C for 2 min, then 45 cycles of [95 °C for 5 s and 60 °C for 30 s]. An internal validation showed that the sensitivity of the PCR was comparable to larval cultures (290 samples were tested by both PCR and larval culturing, showing similar sensitivities for the methods, i.e., 193 samples were positive by PCR and 188 by larval culture).

### 2.5. Data Editing and Statistical Analyses

Horse identification numbers were not available in the data. The name of the horse together with information about the equestrian premise were used to create unique identities. Data editing and statistical analyses were performed in Microsoft Excel (Microsoft^®^ Excel^®^ for Microsoft 365 MSO (Version 2209 Build 16.0.15629.20196) 32-bit) and R (3.6.0). Relationships between categorical variables were evaluated by Chi-Square tests of independence, and differences in mean values were calculated using Student’s *t*-tests.

## 3. Results

Between 1 March 2008 and 30 June 2017, data were obtained from 43,330 faecal samples collected from 26,625 horses on 935 equestrian premises. Yearly, the number of faecal samples ranged from 3240 to 4946 and were submitted from 244 to 324 equestrian premises. The participating equestrian premises were located all over Sweden; however, there was an overrepresentation of premises in central Sweden (the area of Stockholm), where the number of horses sampled in relation to the number of horses registered at the Swedish Board of Agriculture was higher than in the rest of the country [1]. The total number of analysed faecal samples did not vary substantially between the years (Table 3). The number of equestrian premises that requested individual analyses for *S. vulgaris* increased over the years from 2013 when this option was introduced (Table 3).

### 3.1. Occurrence of Strongylus vulgaris

Larval cultures showed that the vast majority of larvae belonged to cyathostomins. The proportion of individuals that tested positive for *S. vulgaris* each year in 2013–2017 varied from 4 to 11%. The proportion of equestrian premises that had at least one horse with *S. vulgaris* increased over the years, with the lowest prevalence in 2008 (6%) and the highest in 2015 (53%) (Figure 1). The proportion of *S. vulgaris*-positive equestrian premises was significantly higher (*p* = 0.006) in 2013–2017 (mean 43%, 95% CI 9.5) compared with 2008–2012 (mean 21%, 95% CI 12.4). The prevalence of *S. vulgaris* was higher among equestrian premises that chose individual analysis of their samples (Figure 2). This difference was significant in 2015 (*p* < 0.001) and 2016 (*p* = 0.015). Among the horses diagnosed with *S. vulgaris*, 75% had not been sampled the previous year.

### 3.2. Strongylus vulgaris in Relation to Egg Shedding and Age

Between 53% and 61% of horses per year had FECs up to 200 EPG. The prevalence of *S. vulgaris* was not correlated with high egg shedding (Figure 3). In contrast, horses with up to 200 EPG were significantly more often positive for *S. vulgaris* than horses with more than 200 EPG (*p* < 0.001). The occurrence of *S. vulgaris* was significantly (*p* < 0.001) higher in horses under 5 years old (10%) than in horses aged 5 years or more (6%).

### 3.3. Patterns of Strongyle Egg Shedding

For the 33,138 submitted samples, the age of the horse was recorded and 90% of those were aged 5 years old or more. The output of eggs varied significantly (*p* < 0.001) between age groups where 2-year-old and 3-year-old horses had the highest EPG values and horses ≥5 years had the lowest (Figure 4). It was also noted that 80% of the total output of eggs originated from 26% of the individuals (Figure 5). Based on both strongyle egg excretion (>200 EPG) and the occurrence of *S. vulgaris* (individual level), it was recommended to deworm 41% (37–45%) of the horses.

Among the 1034 individuals examined in three consecutive years (2014–2016), 62% showed the same result in all three years when the results were categorised as either low egg shedders with ≤200 EPG or high egg shedders with >200 EPG. When using 500 EPG as the threshold for high shedders, 66% remained in the same category.

### 3.4. Non-Strongylid Parasites and Results from the Semi-Quantitative Helminth Egg Detection Method Used in 2017

In 2008–2015, the proportion of *A. perfoliata*-positive equestrian premises varied from 30% (2016) to 47% (2015). In 2017, after the change of method as described previously (see Section 2.3), the proportion of tapeworm-positive equestrian premises increased to 61% (Figure 6). The occurrence of *A. perfoliata* egg-positive individuals varied from 3.4% (2013) to 5.7% (2015), while in 2017, the proportion of tapeworm-positive horses increased significantly (*p* < 0.01) to 9.7% (Figure 6).

Eggs of *Parascaris* spp., eggs of *Strongyloides westeri*, as well as oocysts of *Eimeria leuckarti* were detected in less than 0.1% of the samples. The proportion of horses with *Parascaris* spp. was highest at the age of 1 year (24%, *n* = 620); it then declined with increasing age. Only 0.3% of horses shedding eggs of *Parascaris* spp. were ≥5 years old.

The results from the semi-quantitative method obtained in 2017 are presented in Table 4.

## 4. Discussion

In the mid-1990s, when routine treatments were performed three to four times per year, the occurrence of *S. vulgaris* on Swedish equestrian premises was still 14% [12]. Moreover, an abattoir study showed that as many as 62.7% of the examined horses had slight to severe pathological changes in the cranial mesenteric artery and its main branches [13]. A rise in the occurrence of *S. vulgaris* has been described in Sweden [14] and Denmark [15], while low prevalence figures continue to be recorded in Germany [16,17]. Since no anthelmintic resistance has been reported for *S. vulgaris*, this increase might be partly related to the fact that, for the past 15 years, anthelmintic treatments have often only been based on FECs [14]. In this study, although not representing the whole Swedish horse population, the occurrence of *S. vulgaris* varied over the years, but was markedly higher between 2013 and 2017, compared with the period 2008–2012. This is likely to be related to the fact that, since 2013, the majority of the horses were tested individually, which meant that lower levels of infection could be detected when a larger volume of faeces was analysed from each individual. The data in the present study, as well as from another Swedish laboratory [18], show a considerably lower individual *S. vulgaris* occurrence compared with the study of Tydén et al. [14]. This difference may possibly be explained by the fact that, in the latter study, only 31% of the equestrian premises routinely requested *S. vulgaris* diagnostics. Remarkably, in the present data, most of the horses with *S. vulgaris* had been sampled within the programme for the first time. Thus, it is assumed that newcomers are often responsible for the introduction of the parasite to equestrian premises. This should drive increased awareness of the importance of treating new horses and checking the treatment efficacy before letting them out on grazing areas. Routine treatment of new horses before letting them out on pastures is also recommended by ESCCAP [19].

Unlike the study of Tydén et al. [14], young horses were found to be infected with *S. vulgaris* to a greater extent than older ones, even if this conclusion cannot be drawn for the whole dataset because of incomplete age information. Although it seems logical that immunologically naïve, young horses show a higher occurrence of *S. vulgaris*, it would appear that immunity only reduces pathological injuries, but does not interrupt the life cycle [17].

SVA’s monitoring programme was initiated to help equestrian premises maintain low contamination levels on their pastures, preserve parasitic refugia and minimise the use of anthelmintic drugs. SVA generally recommends the deworming of horses excreting more than 200 EPG and/or are *S. vulgaris* positive and/or shed tapeworm eggs. These data show that a minority of horses in an equestrian premise excrete most of the eggs, which is in line with the “20/80 principle”, i.e., approximately 20% of the horses shed 80% of the eggs [6,20]. Consequently, less than 50% of the horses in the programme are dewormed after each sampling occasion, increasing the chance of slowing down the development and spread of anthelmintic resistance. In the present study, 62% of the horses were recurrent low or high egg-shedders, which is of practical relevance, as the SVA recommends that the former do not need to be sampled in the autumn and the latter often need to be dewormed more frequently [21]. Egg-shedding consistency has also been described in other studies [21].

In groups of adult horses, the SVA does not generally recommend non-specific/pre-emptive anthelmintic treatments for all horses but suggests only selective treatments. However, to reduce the occurrence of *S. vulgaris*, which is still prevalent in Swedish equestrian premises, non-specific treatments are recommended when summer pastures are infected by *S. vulgaris*. In this case, since 2019, we recommend that macrocyclic lactones are administered to all horses in October and March for two consecutive years before returning to the selective treatment programme. The treatment in October is aimed at killing larvae acquired during the grazing season while the second treatment in March is aimed at removing any residual migrating pre-adult stages since it is known that not all the developmental stages are equally susceptible to anthelmintic treatments [22]. To follow up the effect of these treatments and diagnose tapeworm and cyathostomins, it is still recommended that equestrian premises sample their horses before the grazing season. Compulsory treatments of young horses and horses at risk of infection for various parasites are considered in some guidelines [7,23]. The national recommendation is to deworm all foals against *Parascaris* spp. (fenbendazole at 8–10 weeks of age and 16–18 weeks of age) and strongyles (macrocyclic lactones in the autumn). From the second year of life, it is suggested that faecal analyses should be included as part of the parasite control programme [24].

As expected, the proportion of tapeworm-positive equestrian premises and individuals increased significantly after shifting to a more sensitive flotation method. This is in accordance with a Danish study [25], where, depending on the method used, 3.3–8.7% of the samples were positive. Compared with a recent German study [26], however, the occurrence of tapeworms was considerably higher in the present study. One possible explanation for this could be that 30 g of faeces was used here, whereas only 15 g was used in the German study. Considering that post-mortem studies in Spain and Sweden have shown the level of tapeworm infection to be higher in the autumn and winter than in the spring and summer [27,28], one possible improvement to the programme could be to recommend tapeworm diagnostics in the autumn as well, especially in equestrian premises where tapeworms have previously been detected.

## 5. Conclusions

Sweden’s restrictive approach to anthelmintic treatment has encouraged equestrian premises to adopt selective anthelmintic treatment as one of the tools for parasite control. This study shows that the recommended anthelmintic interventions can be greatly reduced when they are based on coprological analyses. The constant presence of a potentially lethal parasite as *S. vulgaris* is somewhat worrying and the effect of recommendations of blind treatments for newcomers and extra treatments in infected equestrian premises will be monitored. Moreover, the monitoring programme itself has continuously generated updated figures on the occurrence of equine intestinal parasites and made equestrian premises aware of the effectiveness of their efforts to manage parasites. In addition to autumnal tapeworm diagnostics, an annual faecal egg count reduction test (FECRT) could be one way to further improve the monitoring programme.

## Figures and Tables

**Figure 1 animals-13-02741-f001:**
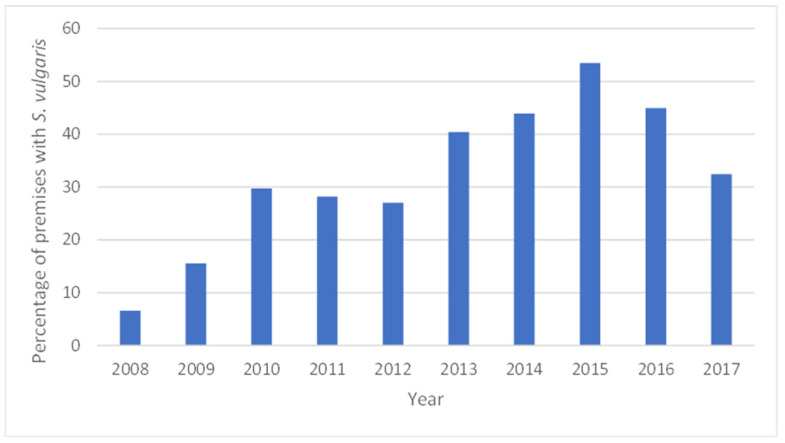
Equestrian premises where *Strongylus vulgaris* was found to be present (at least one positive horse), expressed in percentage per year.

**Figure 2 animals-13-02741-f002:**
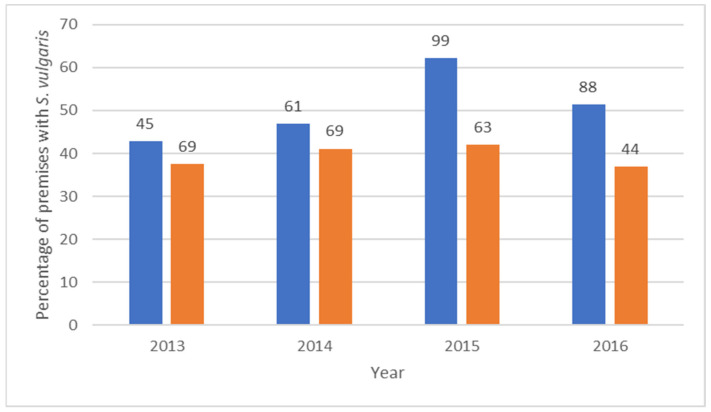
Proportion and number of *Strongylus vulgaris*-positive equestrian premises where larval cultures had been performed on either individual (blue) or pooled samples (orange).

**Figure 3 animals-13-02741-f003:**
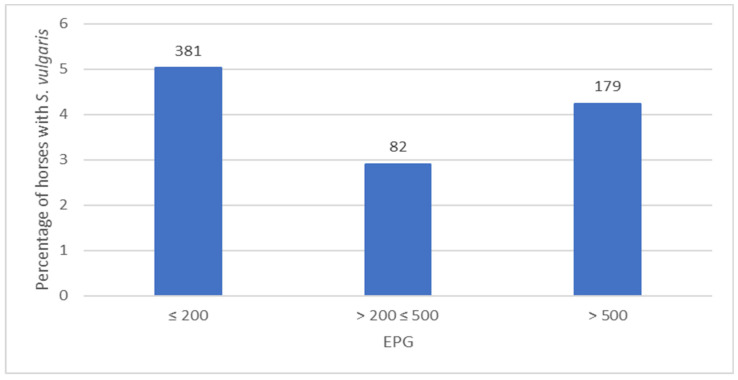
Occurrence of *Strongylus vulgaris* in 2013–2016 in relation to egg-shedding level expressed as EPG (eggs per gram) range. Number of horses are shown above the bars.

**Figure 4 animals-13-02741-f004:**
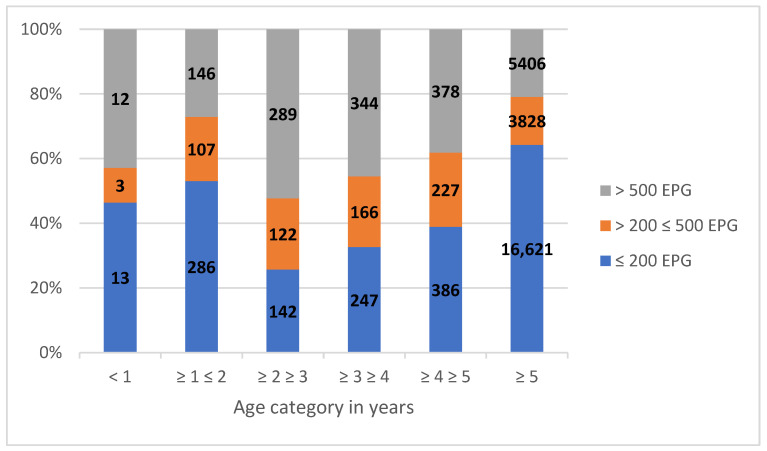
Proportion (*y*-axis) and number (indicated in the bars) of horses in different EPG (eggs per gram) levels per age category between 2009, the first year when age was recorded, and 2016, the last year when McMaster egg counting was performed.

**Figure 5 animals-13-02741-f005:**
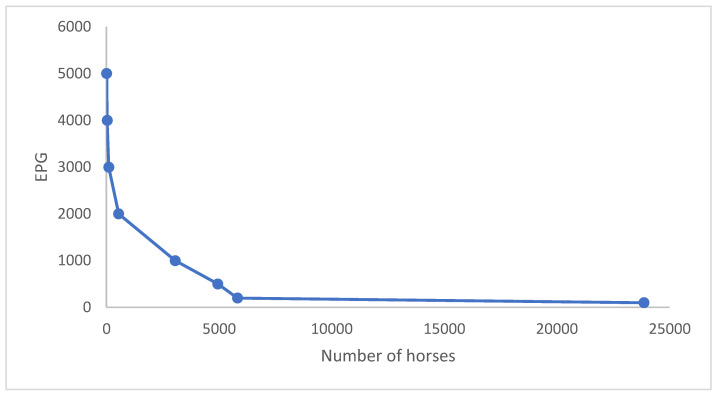
Individual EPG (eggs per gram) for horses that were analysed in 2008–2016. The majority of eggs were shed by a few horses.

**Figure 6 animals-13-02741-f006:**
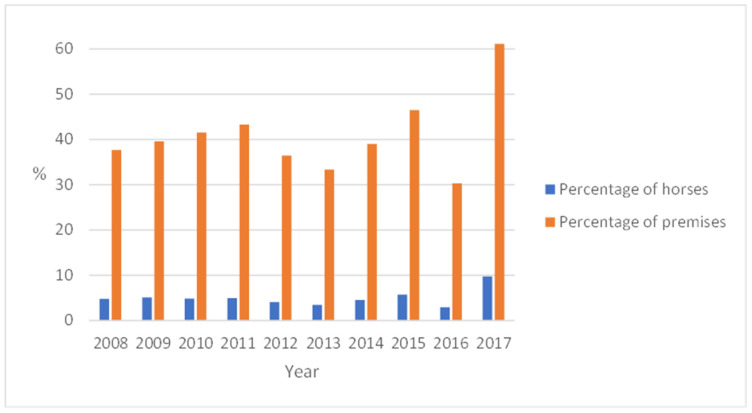
Yearly occurrence of *Anoplocephala perfoliata* eggs in faecal samples. In 2017, samples were analysed by a modified flotation method based on 30 g of faeces, while McMaster on 3 g of faeces was used in previous years.

**Table 1 animals-13-02741-t001:** Overview of methods used in the spring for faecal analyses of horses within the parasite monitoring programme of the National Veterinary Institute (SVA) in different years, between 2008 and 2017.

Method	Year(s)
Eggs: McMaster (3 g faeces) + centrifugation (1× *g*)	2008–2016
Eggs: modified flotation (30 g faeces)	2017
*S. vulgaris*: larval cultivation, pooled	2008–2012
*S. vulgaris*: larval cultivation, individual or pooled (optional)	2013–2016
*S. vulgaris*: individual PCR or pooled larval cultivation (optional)	2016–2017

**Table 2 animals-13-02741-t002:** Assessment of the number of nematode eggs found with the modified flotation method used in 2017. The examination was performed at 100× magnification.

Microscopic Findings	Assessment of Number of Eggs
No eggs	Negative
1–4 eggs under cover glass	Very few
5–10 eggs under cover glass	Low
3–5 eggs per field of view	Moderate
>10 eggs per field of view	Abundant
Most fields of view covered with eggs	Very abundant
Area under cover glass covered with eggs	Mass occurrence

**Table 3 animals-13-02741-t003:** Number of horses and equestrian premises per year included in the parasite monitoring programme and number of equestrian premises that had their samples analysed for *Strongylus vulgaris* either individually or pooled. It should be noted that larval cultures were replaced by real-time PCR in 2017. NA means not applicable.

Year	Number of Horses	Number of Equestrian Premises	*S. vulgaris* Individual Analyses (Premises)	*S. vulgaris* Pooled Analyses (Premises)
2008	3240	244	NA	244
2009	4034	270	NA	270
2010	4477	303	NA	303
2011	4371	284	NA	284
2012	4277	285	NA	285
2013	4320	282	105	184
2014	4368	287	130	168
2015	4695	301	159	150
2016	4602	294	171	124
2017	4946	324	267 (PCR)	62

**Table 4 animals-13-02741-t004:** Results obtained with the modified flotation method (2017).

Assessed Number of Eggs	Strongyle Eggs	*Parascaris* spp. Eggs
Negative	723 (15%)	4833 (98.6%)
Very few	524 (11%)	8 (0.2%)
Low	1171 (24%)	20 (0.4%)
Moderate	1150 (23%)	34 (0.7%)
Abundant	1169 (24%)	6 (0.1%)
Very abundant	164 (3%)	0
Mass occurrence	0	0

## Data Availability

The data that support the findings of this study are available from the National Veterinary Institute (SVA) upon reasonable request.

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
