# Peer review of "Selective Anthelmintic Treatment in Horses in Sweden Based on Coprological Analyses: Ten-Year Results"

_animals, 2023, doi:10.3390/ani13172741_

Round 1

Reviewer 1 Report

Thank you for the opportunity to review this manuscript on selective treatments in horses. Few studies address the long-term impact of selective treatments; hence this research/manuscript fulfills a gap in the literature.

General comments

The study dates are unclear until line 177. Consider clarifying on lines 20,  26 and 104.

Minor comments

Simple summary

Line 16: comma after Sweden

Line 18: comma after study

Line 20: “Faecal samples were analyzed twice per year and, depending on the results, equestrian”

Lines 20-21: this suggests that premises not individual horses were treated

Abstract

Line 28: suggest “in total” vs “totally”

Line 29: with the egg counting method also being a flotation method, this sentence might need clarification

Line 29: consider writing out Strongylus the first time it is used

Introduction

Line 43: consider “larvae often accumulate”

Methods

Line 118: consider specific gravity instead of density

It is unclear regarding methods used at different timepoints. Expanding table 1 to include spring and fall would be useful. Also, for lines 88 and 89, wasn’t intensity of egg shedding based on a fecal egg count? As it reads, it appears that 2 methods were used.

Results

Figures 1-3. It would be easier to read these if the number of horses were listed. For figure 1 this might be difficult and it can be determined from table 2. However, it really is needed for figures 2 and 3.

Figure 5. since from 10000 to 25000 is almost the same, consider shortening this end of the table and expanding for horse numbers that are <5000

Line 243: given that the methods used are not the best for Oxyuris, should this be in the manuscript?

Line 244 and 246: spp or sp for Parascaris? It is spp elsewhere

Discussion

Lines 310-311: this sentence needs to be clarified

Limitations with lack of farm size, number of horses in the study the whole time and aging of a horse in the study over time could be mentioned.

References

Please check the formatting and use of italics for scientific names

These have been indicated in the comments to the author. There are some minor punctuation issues.

Reviewer 2 Report

In this article, the authors describe the results from selective anthelmintic treatment of horses using data from feces for their selective treatment from 2008 to 2017.  They utilized fecal samples from equestrian premises submitted twice per year for 10 years.  A total of about 43 thousand fecal samples were analyzed from 26,625 horses from 935 different locations.  The samples were analyzed with standard egg counting methods.  Also, S. vulgaris was detected using larval culture or PCR.  They found that 4-11% of horses tested positive for S. vulgaris and 3-10% for tapeworm.  They found recurrent high and low egg shedders.  They found that 75% of horses testing positive for S. vulgaris were new to the herd.  They found that S. vulgaris positive premises increased when individual samples vs. pooled samples were used. 

This article, describing treating horses selectively guided by coprological analyses over 10 year is novel and is a good fit for the journal Animals. The emergence of anthelmintic resistance is an enormous problem, and strategies to combat this crisis are desperately needed. Selective use, instead of routine treatment, is one such strategy and this paper further demonstrates how that strategy can be effective.  Although this study doesn’t have some of the features that would be really useful in determining effect like a control group with routine treatments and follow up with the same horses every year using the best technique each year (individual testing vs pooled, PCR, float vs McMaster etc), this paper will be useful for other studies to help guide future selective treatment studies.  Based on their S. vulgaris data they conclude that only 41% of the horses needed to be dewormed.  This reduction in anthelmintic use would absolutely contribute to reducing anthelmintic resistance.  However, the feasibility of testing each animal (or premise) for worms on a consistent basis seems daunting and simply not treating an animal in need seems unethical.  In order to not miss those animals, selective treatment may be unfeasible.  Then again, if some are missed but the drug maintains efficacy maybe overall more sick animals will be able to be treated.  Maybe this is more feasible for animals like horses that “have access to limited grazing areas where helminth eggs and larvae are often accumulated”.

Specific comments suggestions:

Major issues:

1.     Were the samples collected in the Spring only (line 29) or were they done twice per year (line 20)?  Were they collected 2x in the Spring?  Twice per year makes it seem like different seasons.  Then line 85 says “spring and autumn”.  Thus, it seems line 29 needs to be clarified that collections were twice a year but analysis was done once a year.  Then line 87 references spring with egg shedding and line 89 references autumn with FEC.  Egg shedding (which you say was tested in the spring) is tested by FEC (which you say was done in the fall) so overall I’m confused about the timing of the collections and the different analysis used. Maybe you mean in the fall the culturing of S. vulgaris or PCR wasn’t done.  Further clarification on methods: How was the sample prepared/shipped/stored?  Also, if a sample was collected and not tested soon after (ie a half year later), you should mention how the sample was stored longer term.

2.     “SVA internal data” is listed in lines 129 and 146.  Especially for the claim that “the sensitivity of this test was comparable to larval cultures” (lines 145-146) this should be a supplementary figure.

3.     Figure 1 shows a rise in premises with S. vulgaris (although a decrease from 2016 and 2017).  This seems bad, considering the selective treatment would have started in 2010 (presumably because after two years a selective plan was introduced (line 90-91).  Do the authors have a comparison of what the predict routine treatment would have looked like from previously published data?  What changed in 2016-2017 to make it go down?

4.     Figure 2 shows that testing for S. vulgaris on individual horses showed higher amounts than pooled.  Does this mean that pooled testing isn’t a sensitive enough assay?

5.     I would make a figure out of lines 238-241.  Following of the same horses over time is important.

Minor issues:

Check for needed comma use.  For example, in the first paragraph alone, a comma is needed on line 16 after “Sweden”, and on line 20 after “year” and after “results”.

Line 18: define “substantially”

Line 24: when did selective treatment recommendation in Sweden begin?  I see line 59 says 2006 the monitoring program started, so presumably after that.  Again, line 51 says “recently” implemented.  When?

Line 42-43: Citation needed for location of helminths

Line 55-56: confusing that you say “although deworming without prior fecal analysis is still permitted in Sweden” and then follow with “The treatment decision is then based on the results of individual fecal analysis”.  I know you are talking about the first part of the sentence before “although” but you should make this clearer.

Line 137: is that a total of 10 g of feces from four horses, or 40 g total, 10 g from each horse?

Line 202-203: what is the explanation for 75% of the horses diagnosed not having been sampled the year before?  New horses coming into the premises?  Different premises?

Line 310: missing words?  “This…???? since”

Lines 316-317: does this explain the increase from <1 year to >/1/<2 in figure 4?

Just minor changes (like commas)

Reviewer 3 Report

Dear Authors,

Concerning your manuscript Animals-2488853 “Selective anthelmintic treatment in horses in Sweden based on coprological analyses: Ten years’ results” I believe it is an interesting topic and it provides epidemiological data on helminthosis on horse farms in Sweden.

Regarding my reviews and comments, they are as follows:

1. Introduction

I think that what was declared in lines 68-72 "The laboratory methods... is presented in Table 1" should probably be moved to Materials and Methods – Analyses for helminths eggs

Line 52. Please add ref

2. Materials and methods

2.1 The monitoring programme

Line 89-90. In the autumn…carried out. I believe the FEC was done the same way as in the spring. Please standardize or explain the methods applied.

Line 92-94. How many horses needed anthelmintic treatment? was a FECRT performed?

Line 93. Please through the manuscript change epg with EPG

2.3 Analyses for helminth eggs

Line 124: “sugar salt solution”. Please add the density

2.4 Analyses for Strongylus vulgaris

Line 137: “Approximately 10 g of faeces from four individual horse”. I think 10 g for each horse for a total of 40 g of faeces. Please clarify. Also, were coprocultures performed for each positive horse regardless of EPGs? Please add this information.

Results

3.1 Occurrence of Strongylus vulgaris

Line 201. Please change p

3.3 Patterns of strongyle egg shedding

Line 224-226. Approximately 90% of those… (Fig.4) Why approximately? How many horses aged more than 5 years? Considering that the horses have been divided by age classes, I kindly ask to show the number of horses (and the percentage) belonging to each age class. It would also be useful to show in detail the average EPGs for each age class.

Figure 1. suggest adding the percentage values above the bars.

Figure 3. Please report the percentage value on the scale bar

Figure 4. I'm sorry but I don't understand the graph. What does the % on the y-axis mean? The number of horses belonging to each EPG class? Please clarify.

3.4 Prevalence of non-strongylid helminths

In paragraph 2.3, table 2, the eggs observed under have been used to ass the number of eggs. I kindly ask you to report the data obtained also in the results. How many were negative, very few and so on, and for which eggs?
